# Deep phylogeny of cancer drivers and compensatory mutations

Nash D. Rochman[1], Yuri I. Wolf [1] & Eugene V. Koonin [1✉]

Driver mutations (DM) are the genetic impetus for most cancers. The DM are assumed to be deleterious in species evolution, being eliminated by purifying selection unless compensated by other mutations. We present deep phylogenies for 84 cancer driver genes and investigate the prevalence of 434 DM across gene-species trees. The DM are rare in species evolution, and 181 are completely absent, validating their negative fitness effect. The DM are more common in unicellular than in multicellular eukaryotes, suggesting a link between these mutations and cell proliferation control. 18 DM appear as the ancestral state in one or more major clades, including 3 among mammals. We identify within-gene, compensatory mutations for 98 DM and infer likely interactions between the DM and compensatory sites in protein structures. These findings elucidate the evolutionary status of DM and are expected to advance the understanding of the functions and evolution of oncogenes and tumor suppressors.

---

[1] National Center for Biotechnology Information, National Library of Medicine, Bethesda, MD 20894, USA. ✉email: koonin@ncbi.nlm.nih.gov

The rapid decrease in DNA sequencing costs[1] has enabled an extensive survey of the pan-cancer mutational landscape, with the data made publicly available through the landmark projects, COSMIC: the Catalog of Somatic Mutations in Cancer (cancer.sanger.ac.uk/)[2] and The Cancer Genome Atlas (TCGA) Research Network (cancer.gov/tcga)[3]. Supported by these advances, a large body of work now exists separating cancer 'driver' genes as well as specific 'driver' mutations that are thought to mold the tumor phenotype from the larger list of 'passenger' mutations with no known functional impact in isolation[4–7]. Whereas classical mutational time series have been proposed to underpin tumorigenesis for more than three decades[8], epistasis among cancer driver genes remains actively researched[9–12]. Recent work aims to construct a generalizable framework for understanding the order in which drivers appear[13,14] as well as the role of passenger accumulation[15,16] in tumor evolution. Compensatory mutations, i.e., strong epistatic interactions often producing the opposite effect in concert from that of each constitutive individual mutation, among driver genes have been shown to confer drug resistance to tumors[17–19]. Furthermore, multiple modes of somatic mosaicism have been documented[20] where reversion or de novo compensatory mutations mitigate the effects of a deleterious germline variant[21,22]. Compensatory mutations for drivers have been engineered in vitro yielding both a method for validating driver status and general information about protein structure and function[23–27]. However, few studies (for example, the identification of mutually compensatory mutations in TP53) provide examples of such compensatory pairs of mutations in orthologs of cancer driver genes from other species[28].

Mapping compensatory mutations onto protein crystal structures and validating the corresponding interactions between amino acid residues and their effects experimentally is, arguably, the optimal approach to elucidate important functional features of the target protein and can produce unambiguous results. Crystal structures, however, are often challenging to resolve, and despite the remarkable progress in the field, it will require a major, long-term effort to obtain structures of many, particularly membrane, proteins[29–31]. Although uncommon, driver mutation states are present in orthologs of cancer driver genes throughout the species tree, and exploration of the evolutionary landscape of co-occurrence between drivers and other mutations in these orthologs is likely to bring to light candidate driver compensators which could motivate crystallographic validation and functional studies. From the standpoint of evolutionary biology, cancer drivers appear to be of special interest because these mutations are a direct manifestation of the fundamental evolutionary conflict between the 'interests' of individual cells in maximizing proliferation and those of multicellular organisms for which it is essential to keep cell division in check. The evolutionary status of driver mutations outside vertebrates has not been studied in detail, and basic questions stemming from this evolutionary conundrum remain unanswered. How does the likelihood of observing a driver state depend on the evolutionary distance from mammals? Are drivers universally avoided or are they more commonly observed in unicellular life forms compared to multicellular organisms as one could suspect given their effect on cell proliferation? Are drivers generally deleterious, and accordingly, when drivers are present in other species, is their detrimental effect compensated by other mutations? Here we examine deep phylogenies of cancer driver genes for the occurrence of driver states and potential compensatory mutations to shed light on these basic questions. We expect that our list of likely compensatory mutations provides direction for further experimental validation.

## Results

**DM prevalence site depth, not tree distance-dependent.** In this work, we present deep phylogenies for a set of 84 genes (Supplementary Data 1, 15) identified as cancer drivers in multiple tissues with high confidence[4]. From this complete set of established, non-tissue-specific genes, two genes were excluded: KMT2C/D due to an impractically large number of paralogs and HLA-A which resides in the same MSA as HLA-B. We establish the prevalence of 434 driver mutations across the gene-specific species trees constructed from protein multiple sequence alignments (MSA). Only missense mutations were considered, to allow for clear identification in the MSA. The MSA were constructed to be as deep and phylogenetically inclusive as possible (see "Methods" for details), and long paralogous branches were manually removed, resulting in alignments that typically contained no more than three sequences per species excluding plants for which greater numbers of co-orthologs were included (Supplementary Fig. 1). Approximately half of the MSA include orthologs from fungi, plants, or both, and six include multiple prokaryotes, whereas the rest are exclusively metazoan (Fig. 1 and Supplementary Fig. 2). For all MSA, the representation of eukaryotic species is largely uniform across the major clades, especially, for plants (Fig. 1). The exception is unicellular eukaryotes (protists) among which only a minority possesses an ortholog of any given driver gene. It might be tempting to rationalize this observation by concluding that driver genes, mostly, evolved and persist in multicellular eukaryotes, but caution is due because of the insufficient and uneven sampling of the numerous protist lineages (Supplementary Fig. 3). For instance, the uneven representation of cancer drivers in protists could be due to gene loss in parasites. Additional genome sequencing of a broad array of protists is needed for a robust assessment of the association (or lack thereof) of the evolutionary conservation of cancer driver genes and multicellularity.

Seeking to establish a conservative list of drivers to investigate for each gene, we calculated a measure of conservation, homogeneity (see "Methods" for details), among vertebrates in all sites and for neighborhoods ($+/-3$ sites) that harbor mutations from the COSMIC[2] database. Each mutation (driver candidate), excluding common human polymorphisms (labeled SNP in COSMIC), was assigned a rank ($1+$ the number of distinct mutations observed more frequently than the given mutation). Alternatively, mutations were ranked by their frequency in tumors (Supplementary Figs. 4 and 5). Top-ranked driver candidates are predominantly found in highly conserved regions of the respective proteins, and both site and neighborhood homogeneity decrease with increasing rank (Fig. 2a). As could have been expected, top driver candidates are uncommon in other species, such that the COSMIC frequency is inversely correlated with the frequency in orthologs across species: leaf-weighted frequency (see "Methods" for details) among species increases with the rank across all major clades. For the lowest-ranked driver candidates (those predominantly observed in only one tumor and likely to be effectively random), the frequency of presence among distant eukaryotes (protists, fungi, and plants) approaches 5%, roughly, the probability of observing a random residue in an arbitrary site, 1/20 (Fig. 2b).

Given the dramatically different contexts of species and tumor evolution, one might surmise that there should be no relationship between the frequency of driver states in tumors and in species, which is in direct contradiction to our findings. A driver mutation appearing in the evolutionary record of multicellular species preceded by a compensatory mutation is a neutral event whereas that same mutation appearing in a tumor is under positive selection. However, we provide evidence below that not all drivers

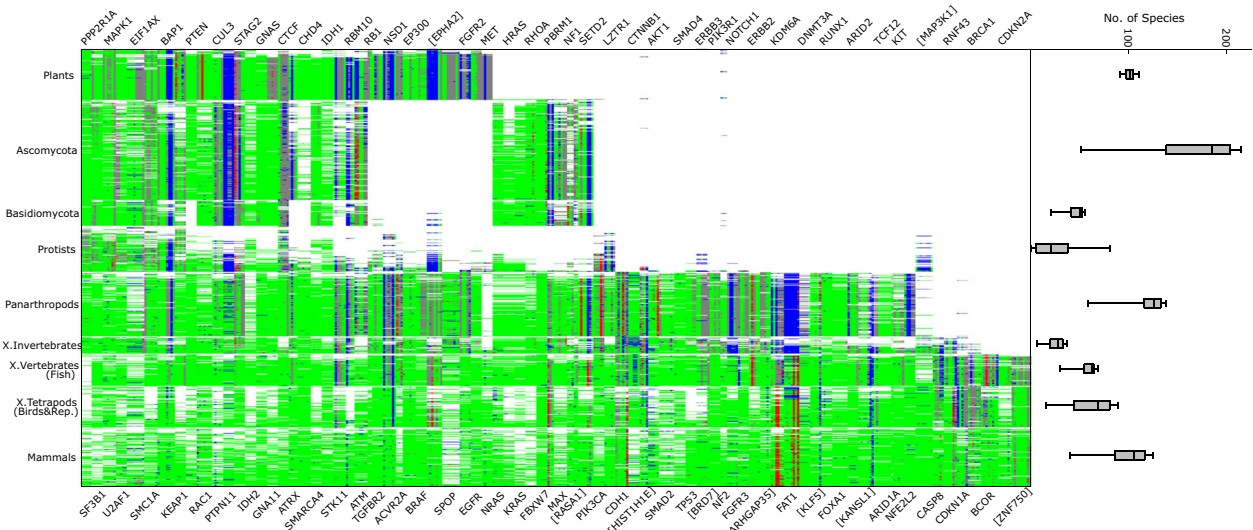

**Fig. 1 Deep phylogenies of cancer driver genes.** Each row represents one species, each column one driver. Sites harboring multiple drivers appear multiple times. Colors correspond to mode residues over all sequences from each species in each site: white, absent from MSA; blue, gap; green, human reference residue; red, driver; gray, any other residue. Species are ordered by taxonomy, and within labeled clades, by appearance within # of MSA (e.g., a plant found to have orthologs in 310 drivers would occupy a row below another plant with 103). Sites are ordered by the phylogenetic depth of the respective genes. Only eukaryotes are shown, for prokaryotes see Supplementary Fig. 2. Rows are followed by box plots of the number of species within each clade observed across MSA where the given clade is represented. Whiskers are at 2/98%.

are deleterious across all multicellular species and thus are not always compensated. Also, some drivers are likely to be only weakly deleterious, so that, even if eventually compensated, they might precede the compensatory mutation in the course of evolution. Overall, these findings are compatible with our observation that mutations less commonly observed in tumors are more likely to be tolerable in multicellular organisms and thus are more frequently fixed in the course of evolution.

These observations motivated us to define rank thresholds for each gene and select driver candidates above these thresholds for further consideration. All but three genes (*HLA-A/B*, SNP dominated, and *APC*, nonsense dominated) contain at least one missense mutation within rank 20 or lower (Fig. 2c). We selected up to 9 top driver candidates per gene (Supplementary Fig. 6) that (1) have rank below 20 and (2) are observed less frequently than at most 5 distinct missense mutations. Selection under these criteria yielded a set of 434 driver mutations. Although many more sophisticated methods for reliable driver identification have been developed[32], we did not restrict our list of drivers in any other way to guarantee a comprehensive survey of the most common missense mutations observed in cancer across the species tree.

On the whole, driver states in this ensemble are observed less frequently among vertebrates than other substitutions relative to the consensus residue (Fig. 2d) indicating that, even when not explicitly demonstrated to be deleterious, these mutations are widely avoided during animal evolution. The driver mutations mainly reside in highly conserved sites, which is compatible with the functional importance of the respective residues, such that mutations exert a deleterious effect. As could be expected, driver states are most strongly avoided in vertebrates but, perhaps surprisingly, their frequency differs little between invertebrates, fungi, and plants. By contrast, drivers are significantly more prevalent among unicellular eukaryotes (Fig. 2e; see the legend for *p*-values). Although too few driver genes have orthologs in prokaryotes to demonstrate statistical significance, driver states were found to be rare even in prokaryotes. This partly results from the fact that prokaryotic orthologs are detectable only for highly conserved proteins, and in general, deeply conserved sites

(those with confidently detected counterpart sites in the fungal and/or plant orthologs) show a higher homogeneity in vertebrates than 'shallow' (exclusively metazoan) sites (Fig. 2f, left). Surprisingly, even when the comparison was limited to highly homogenous sites, driver states were less frequently identified in deep than in shallow sites (Fig. 2f, right).

Thus, in some ways, the frequency distribution of driver states across species matches the expectation. Substitutions resulting in driver states are uncommon, and the frequency distribution sharply decays (Fig. 2g), with 181 of the 434 drivers being universally avoided. However, in those MSA sites that do include driver states, their frequency is uniform (when averaged across all sites) among invertebrates, fungi, and plants (Fig. 2e), indicating a near-constant deleterious effect of the driver substitutions across the major branches of the species tree including distant ones. Thus, the probability of observing a cancer driver state in any species depends more strongly on the phylogenetic depth of the respective site than on the class or even kingdom where the species belongs. In other words, the deleterious effect of a driver state depends primarily on the conservation and hence shared functional importance of the given site within a gene, which are conceivably stable through long evolutionary spans, rather than on the evolutionary distance of a clade from mammals.

**Some DM are ancestral states in major clades.** Despite the overall rarity of the driver states across species, 215 of them were found to be the mode residue in the respective site in at least one species, and 18 are dominant or predicted ancestral states in major clades (Fig. 3; Supplementary Fig. 9; Supplementary Data 2–4)[33–37]. For each of these 215 drivers, we identified the "target clade" being either the largest taxonomic group in which more than half of the species harbor the driver state or the smallest taxonomic group containing more than 90% of all the species harboring the driver state, whichever is smaller. In other words, we found the largest group where the driver is common unless a subgroup can be identified which covers almost all the instances of that driver across the tree. (Supplementary Data 2).

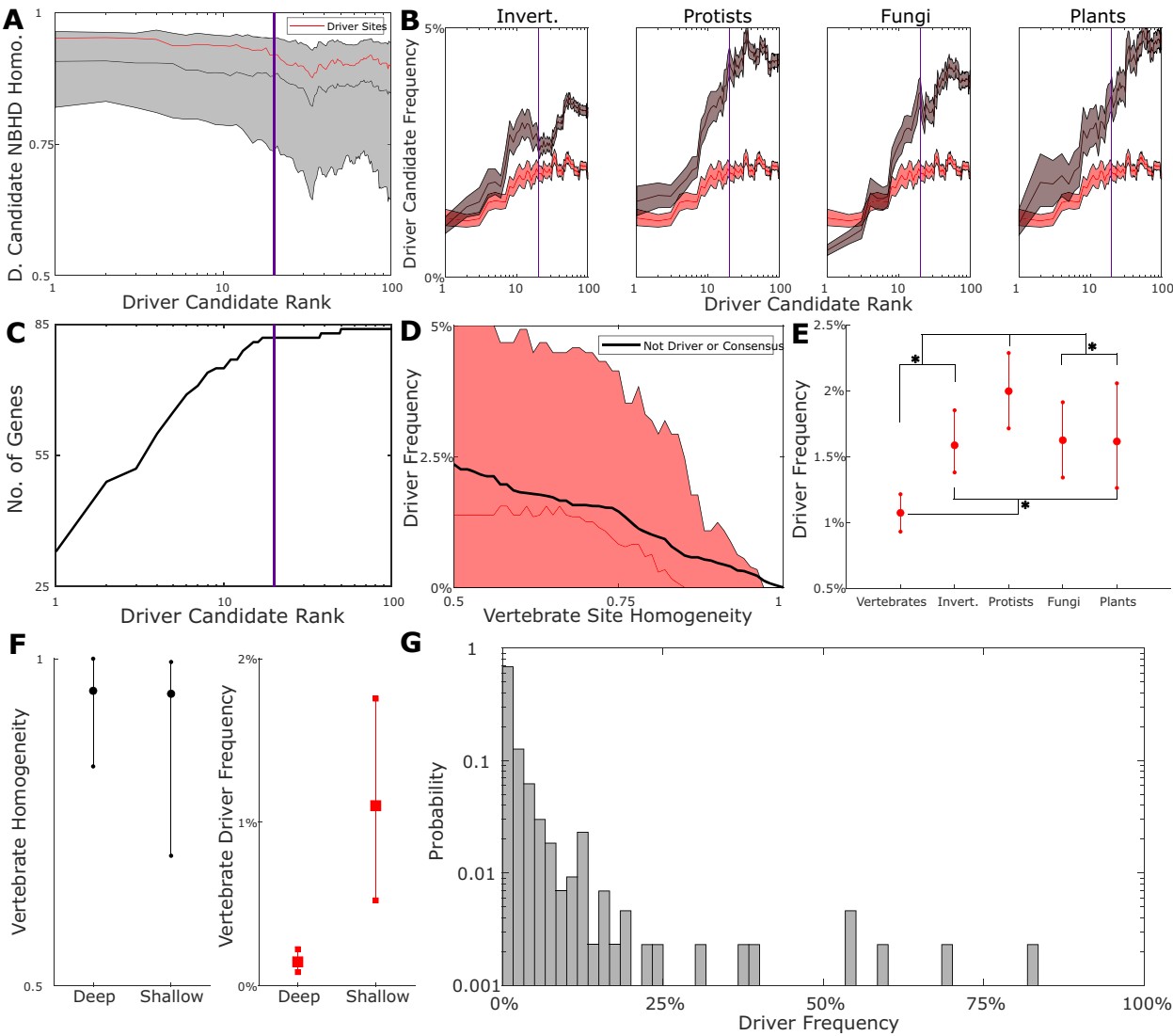

**Fig. 2 Prevalence of cancer driver states depends on site depth. a** Median homogeneity of driver candidate neighborhoods (+/−3 sites) in vertebrates and sites logarithmically binned by rank. The 25th–75th percentiles shaded. Vertical line shows the rank threshold of 20. Vertebrate data repeated in red. Invertebrate, protist, fungal, and plant data displayed, left to right, in gray. **b** Median, 25th, and 75th percentiles of the mean (bootstrap with replacement 1000 times) driver candidate frequency across major clades logarithmically binned by rank. Vertical line, rank 20. **c** #Driver genes (84 total) with at least 1 missense mutation of specified rank). Vertical line, rank 20. **d** Median driver frequency and non-driver substitution (relative to consensus) frequency binned by vertebrate site homogeneity. Each bin contains nearest 20% of sites. For each site, mean frequency of all driver/non-driver substitutions is taken and each site appears at most once in each bin. **e** Median, 25th, and 75th percentiles of mean (bootstrap with replacement 1000-fold) driver frequency across major clades. All distributions are suitably normal ($p < 0.01$, Anderson Darling) with the following significant ($p < 0.05$) pairs according to a two-sample $t$-test: vertebrates/all ($p < 1e−175$), protists/all ($p < 1e−20$), invertebrates/plants ($p < 1e−6$), and fungi/plants ($p < 1e−9$); asterisks denote $p < 1e−20$. The minimum/maximum of each distribution are as follows: 0.43%/1.85%, 0.78%/3%, 0.92%/3.74%, 0.62%/3.18%, 0.4%/4.75%. **f** Left: Median, 25th, 75th percentiles of "deep" (MSA containing fungi or plants) vs "shallow" vertebrate site homogeneity. Right: Vertebrate driver frequency among deep vs. shallow sites exceeding homogeneity 0.95. The minimum/maximum for each distribution are as follows: Left: 0.4/1, 0/1. Right: 0.002%/0.6%, 0.05%/4.4%. **g** Histogram of drivers binned by species frequency.

When multiple paralogous sequences are present in an MSA for a single animal, fungal, or prokaryotic species, driver states are typically found in all or none of those sequences; by contrast, among plants, protists, and to a much lesser extent fish, the driver state is more likely to be a minority residue when present, possibly, due to the typically larger number of co-orthologs (Fig. 4a). Although half of these cases are observed within vertebrates, it has to be emphasized that most of the harboring sites are evolutionarily shallow, i.e., exclusive to metazoa or vertebrates, and so, as shown above, are expected to demonstrate a high driver frequency compared to deeper sites including fungi or plants. For

such drivers in shallow sites, frequencies among vertebrates and invertebrates are poorly correlated, whereas for drivers in deep sites, frequencies among metazoa and plants/fungi show a stronger correlation (Fig. 4b, left and right, respectively). This observation indicates that, in general, drivers permissible in a given clade are no more likely than average to be permissible in any other clade. Thus, the results match the expectation that drivers present in species evolution are compensated by other mutations and that these compensatory mutations are rare. If a compensatory mutation appears deep in the tree, the driver it compensates is likely to be permissible among disparate taxa.

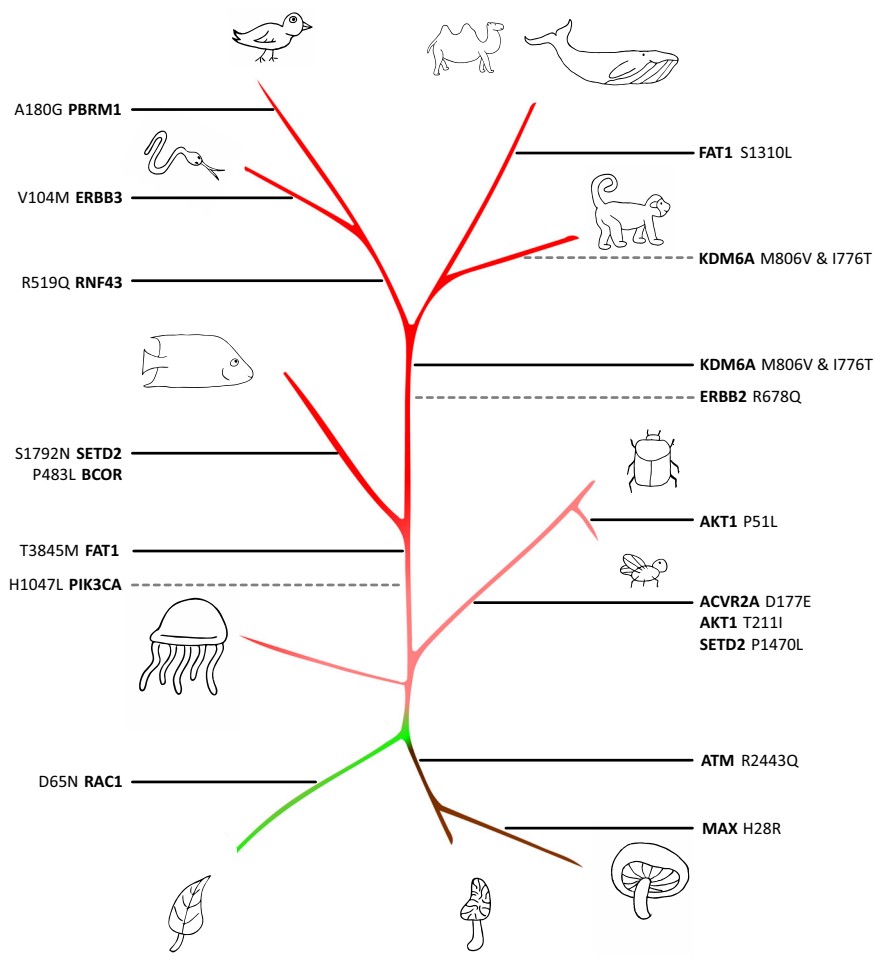

**Fig. 3 The species tree.** This cartoon representation of the tree highlights the emergence (solid, black line) and disappearance (dashed, gray line) of dominant/ancestral driver residues across major clades. The cartoons were generated using Autotracer.org[37].

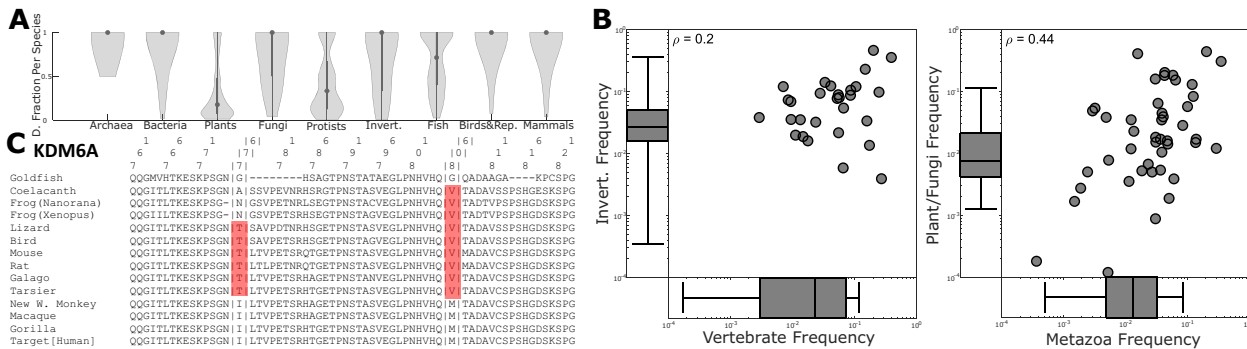

**Fig. 4 Select drivers are ancestral in major clades. a** Violin plots over major clades displaying the driver frequency among all sequences of each species where at least one sequence from that species harbors a driver. Median, 25th, and 75th percentiles of the species shown. Species that harbor multiple drivers are represented multiple times. **b** Left: frequencies of drivers harbored in "shallow" (MSA do not contain fungi or plants) sites among invertebrates vs. vertebrates and observed in both taxa. Right: frequencies of drivers harbored in "deep" sites among plants or fungi vs. metazoa and observed in both taxa. *L&R*, box plot of driver frequencies observed in one but not both respective taxa. Whiskers are at 2/98%. **c** Reduced MSA for driver gene *KDM6A* with drivers I776T and M806V highlighted.

Most of the putative ancestral drivers are harbored in sites within conserved domains or regions of known function such that substitutions in these sites can be expected to exert deleterious effects. In particular, *PIK3CA* H1047L, *ATM* R2443Q, and *ERBB3* V104M are well-documented substitutions that are found in both cancer and hereditary disease[33]. However, when observed in distant orthologs, substitutions relative to the human reference

can not only be benign but essential. For example, *RAC1* GTPase is homologous to plant G proteins in the *ROP* family, and the D65N driver substitution, which is ancestral to plants, has been shown to be important for substrate recognition[35], providing an example of a well-conserved site, in a conserved neighborhood, with different functions across the tree of life. Perhaps, the most remarkable distribution of any driver state across the tree is the

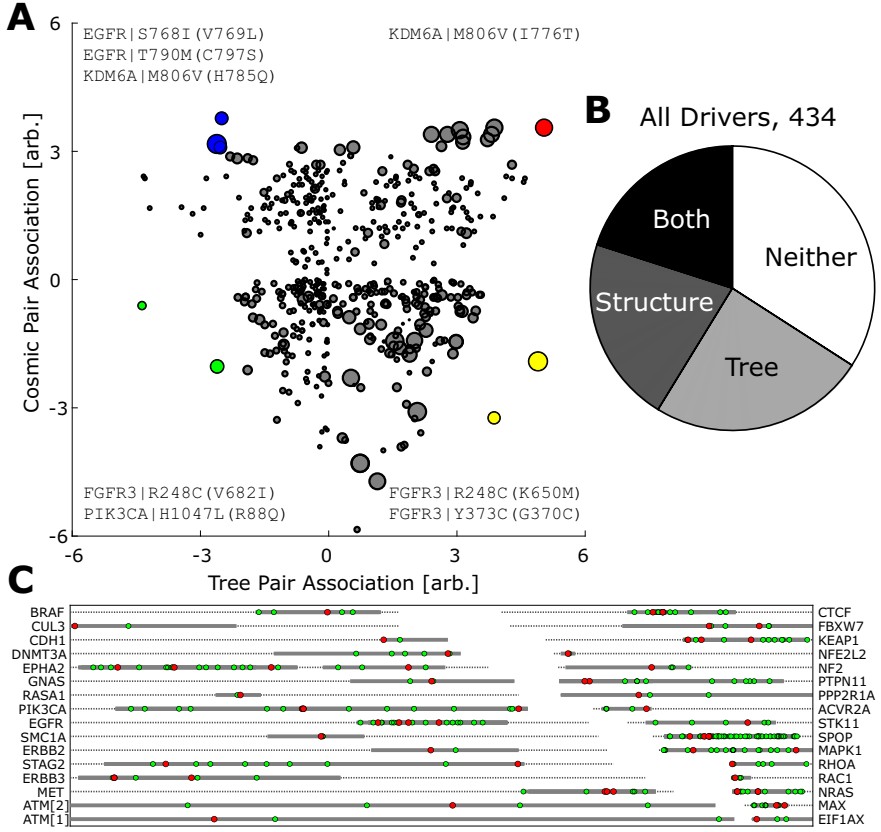

**Fig. 5 Structures and associations among of cancer drivers. a** Association score, indicating the degree to which the co-occurrence frequency of a mutation-pair deviates from expectation if independent (+/−, more/less frequent) for select mutation pairs with large association scores in both COSMIC and the MSA. COSMIC association vs MSA association. Outliers in each quadrant are colored and listed. Bubble size scales with either the expected or observed number of pairs, whichever is smaller, across quantile normalized ex/obs distributions for both COSMIC and the MSA. See "Methods" for details. **b** Pie chart indicating the fraction of drivers with both a "good" structure (whole score >1) and at least one harboring species in the tree (20%), structure only (21%), tree only (35%), and neither (24%). **c** Coverage of driver/compensatory ensemble neighborhoods with structures scored (Local Score) 1.9 or greater. Dashed line, length of protein; gray band, PDB structure; red dots, driver location; green dots, compensator location. *ATM* is resolved in a single structure[75] and split for display purposes.

nearly identical presentation of M806V and I776T in the lysine demethylase *KDM6A* (Fig. 4c). These drivers are two of the most frequently observed mutations in the COSMIC database for *KDM6A*, with none more frequent than I776T and only three missense mutations observed more frequently than M806V. The neighborhood conservation around these sites is strong among tetrapods, less so among fish, and negligible among invertebrates. The residue 806V appears to have been fixed shortly before 776T at the base of tetrapods and amniotes, respectively, and both disappear at the base of monkeys. To our knowledge, the structure or function of this neighborhood in *KDM6A* has not been studied. Nevertheless, this is a notable case of the unusual prevalence of two nearby drivers among most vertebrates, with a concomitant replacement in monkeys suggesting a functionally important interaction between these sites.

Taken together, these findings indicate that the deleterious effects of cancer drivers, although common, are far from insurmountable in the course of evolution, suggesting widespread compensatory effects of accompanying mutations. Thus, we next sought to identify such compensatory mutations for as many drivers as possible.

**Phylogenetic, structural evidence of compensatory mutations.** Before focusing on ensembles of explicitly compensatory mutations for drivers, we reviewed pairs of mutations that are strongly associated (see "Methods" for details) in both the COSMIC database and the MSA of driver genes (Fig. 5a; Supplementary Data 5–10)[33,34,38,39]. Pairs with positive (observed as a pair significantly more frequently than expected in both datasets) or negative (observed significantly less frequently than expected) associations imply similar functional interaction of the sites in both the tumor and across the harboring species. The pair M806V and I776T in *KDM6A* is a striking example of a positive association. In principle, one could expect opposite associations in tumors and among species because adaptations beneficial to a tumor are almost always deleterious for the organism and would be expected to be deleterious to (at least) all multicellular species. Conversely, many adaptations beneficial to multicellular species (e.g., cell-cycle checkpoints) directly stymie tumorigenesis. Contrary to this expectation, in our analysis, we observed many pairs of mutations with either positive or negative associations (Fig. 5a). It remains to be investigated and understood why, in many cases, although the driver is deleterious to the multicellular organism and essential for tumorigenesis, the same compensator seems to provide a selective advantage at the levels of both organisms and tumor cells.

We adopted an approach for detecting compensatory mutations that is much more restrictive than pairwise association. For each driver, we attempted to identify a 'compensatory ensemble' composed of one or more compensatory mutations (or simply, 'compensators') such that all sequences containing the driver state

also contain a member of this ensemble. Although this approach minimizes the chance that a 'compensatory' ensemble is constructed for a driver additionally present in an arbitrary leaf of the tree due to sequencing error, our analysis suggests this does not pose a problem for the dataset analyzed here (see "Methods" for details). To be considered a candidate compensator, a mutation must be predicted to predate the given driver based on the ancestral state reconstruction from the phylogenetic tree. The ensemble must have a low probability of independent co-occurrence with the driver and members of the ensemble appearing at least twice independently, at two nodes in the tree. Each individual mutation may only occur in at most one ensemble and the ensembles were constructed so as to minimize the probability of independent co-occurrence (see "Methods" for details). Few of the mutations represented in these compensatory ensembles are frequently observed in the COSMIC database (Supplementary Data 11). This is largely due to the fact that top COSMIC hits are rarely observed in other species, but even frequent mutations, such as *KDM6A*, M806V, and I776T, would not enter into consideration because at times, 806V appears in the MSA without 776T despite the presentation of this pair being suggestive of mutually compensatory function in mammals.

For the top candidate ensembles, we searched for available structures in the Protein Data Bank (PDB, rcsb.org/)[40] which could support or refute interaction. Although driver neighborhoods are more likely to be structurally resolved than arbitrary regions of a protein, many driver neighborhoods remain unresolved (Fig. 5b) and the majority of those resolved are not entire proteins but rather individual domains (Supplementary Figs. 7 and 8). Compensators, or other associated sites, may be far from the driver in the sequence but close in the structure, so that, when only individual domains of the respective proteins are structurally resolved, these relationships might remain hidden. Without a structure covering all associated sites, few paths for further workup are available. Nevertheless, we identified five driver-compensatory mutations (compensatory ensemble of size 1) pairs with enough phylogenetic evidence to warrant additional consideration (Supplementary Data 12 and 13). In particular, driver E119D, and the proposed compensatory mutation S398N in *RBM10* (RNA-binding Motif 10) is noteworthy as the driver residue is only present in lower mammals, while the compensator transitions out at the base of primates. (Supplementary Fig. 9). Altogether, we identified 32 driver/compensatory ensemble neighborhoods that were fully covered by resolved protein structures (Fig. 5c), and below we focus on 9 characteristic examples of these structures.

Despite the strict criteria imposed, compensatory ensembles for 98 drivers, including the 5 mentioned above, were derived containing 1 (12%), 2 (25%), 3 (21%), and >3 (42%) mutations (Supplementary Data 12–14)[33,34,41,42]. In particular, we highlight 9 notable structures (Fig. 6 and Supplementary Figs. 10–18[43]; see Supplementary Data 12 for details). Three cases present the canonical picture of a compensatory mutation: each compensatory ensemble is of size 1 and a mechanism for the compensator to directly interact with the driver residue and counteract the effect of the driver is readily apparent in the structure (Fig. 6a). In *ERBB2* (erb-b2 receptor tyrosine kinase 2; *HER2*)[44,45], the addition of a methyl group in the driver V842I appears to be balanced by the loss of a methyl group in the compensator T900S, conceivably, preventing the residue in site 900 from further (pathologically) interacting with the next adjacent residue. In *BRAF* (B-Raf proto-oncogene, serine/threonine kinase)[46], the change from a positive to a negative charge in the driver K601E is counterbalanced by the change from a neutral residue to a positively charged one in the compensator, F635R. In *EPHA2* (EPH receptor A2, ephrin receptor)[47], the driver R244H is a

substitute from a strong base to a weak base, and the compensator T225K is a substitute from a neutral side chain to a moderate base, thus balancing local protonation.

Four cases involve modification of a small molecule binding pocket or protein–protein interaction interface (Fig. 6b). In *NRAS* (NRAS proto-oncogene, GTPase)[48], drivers and compensators all appear to modify the binding pocket. In *GNAS* (GNAS complex locus)[49,50], driver R844H likely promotes $Mg^{2+}$ coordination and prevents dissociation whereas the nearby compensator S848P could introduce a kink, opening the pocket and promoting dissociation. In *RAC1* (Rac family small GTPase 1)[51], the driver and compensators all appear to be involved in modifying the binding, cleavage, and release of GDP. In *KEAP1* (kelch like ECH associated protein (1)[52,53], driver G333C and compensators S508P and R554K appear at the binding interface of *KEAP1* and an engineered peptide shown to inhibit the interaction with *NRF2* (NFE2L2; nuclear factor, erythroid 2-like (2). It has been shown that G333C mutants of *KEAP1* are unable to repress *NRF2* activity[40], further demonstrating the functional importance of these sites. S508P potentially balances the increased size of the driver substitution by introducing a kink in the structure and opening up the geometry. R554K could also sterically balance the driver through a slight decrease in size, in addition to modifying the protonation of the interface. In *CTCF* (CCCTC-binding factor, zinc finger containing; Fig. 6c)[54], the compensation mechanism is likely to involve a stabilizing aromatic interaction between the driver S354F and compensator K367Y. Finally, in *PIK3CA* (phosphatidylinositol-4,5-bisphosphate 3-kinase catalytic subunit alpha; Fig. 6c)[55], the multiple drivers and compensators are located in two exterior neighborhoods of the large structure potentially important for protein–protein interaction, suggesting a more complex compensatory mechanism.

## Discussion

The definition of driver mutations, that is, mutations that promote tumorigenesis, implies that these mutations reflect the trade-off between the selection for maximum cell proliferation and selection for cell-cycle control that is essential for multi-cellular life forms. Thus, at least in principle, the study of the evolution of driver states could shed light on the fundamental aspects of the evolution of multicellularity. In this work, we analyze deep multisequence alignments for a representative ensemble of cancer driver genes and explore the appearance and distribution of driver mutations throughout species evolution. This analysis allows us to broadly assess the fitness effects of driver mutations across varying evolutionary spans. In general, driver states are strongly avoided such that almost half of the drivers included in this study are not detected in any of the available orthologs of the driver genes. Thus, the majority, if not all, driver mutations have a negative organismal fitness effect, even in unicellular life forms and those multicellular organisms that are not subject to cancer, such as plants and fungi. In that regard, one has to keep in mind that cancer cell proliferation drastically differs from normal cell division in that tumorigenesis involves various forms of genome instability including aneuploidy[56,57].

Surprisingly, the distribution of drivers is largely non-specific with respect to taxa, and driver states appear to be roughly equally avoided among invertebrates, fungi, and plants. In other words, the prevalence of driver states does not strongly depend on the evolutionary distance of a taxon from mammals. This observation motivates the hypothesis that missense mutations identified as pathological in mammalian or metazoan species outside the context of cancer are widely avoided in general. We identified too few alignable orthologs among prokaryotes for

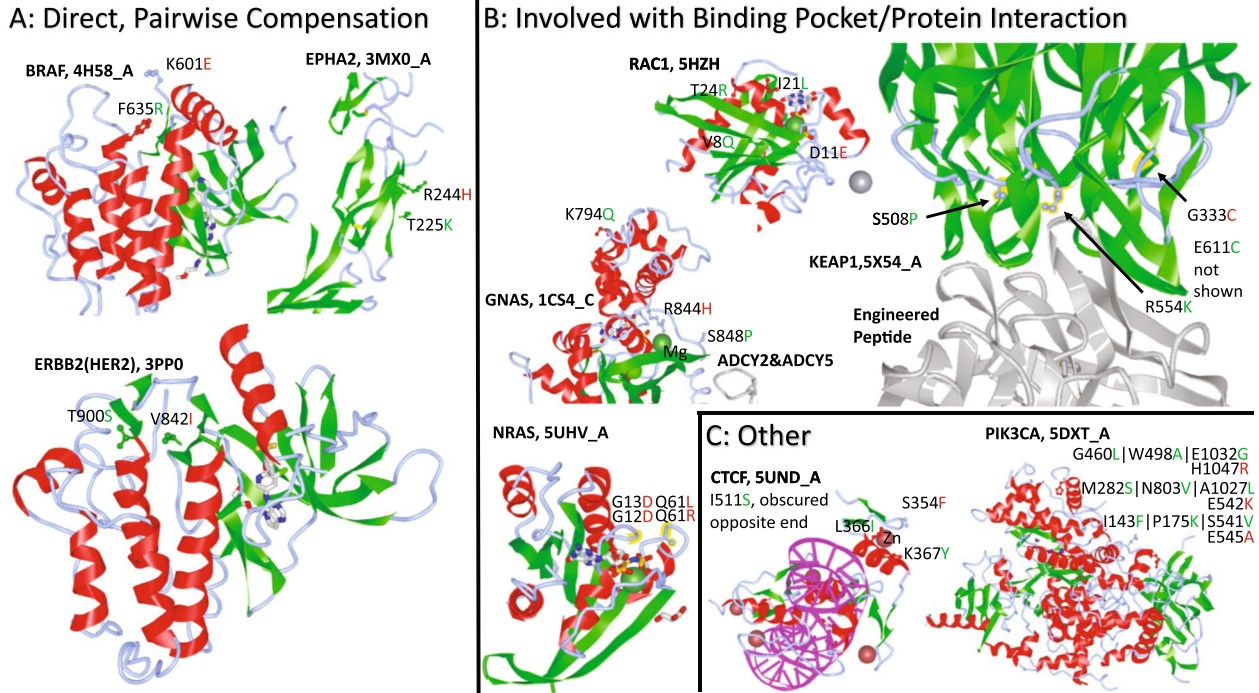

**Fig. 6 Compensatory mutations for cancer drivers. a** Examples of direct, pairwise compensation where there is a single compensatory mutation which likely balances/counteracts the change induced by the driver. **b** Examples of compensatory ensembles which are likely to play a role in modifying small molecule binding pockets or protein interactions to offset the presence of the driver. (In *NRAS*, compensators not highlighted in structure to avoid clutter.) **c** Other examples. In *CTCF*, an aromatic interaction between the compensator and the driver is predicted to stabilize the protein. In *PIK3CA* (compensators not highlighted in structure to avoid clutter), localization of multiple drivers and compensators to two exterior neighborhoods of the large structure potentially important for protein interaction.

robust statistical analysis, but some drivers are completely avoided even in this group. Notably, however, drivers are more common among protists, and driver gene distribution appears to be more heterogenous among the unicellular than among multicellular eukaryotes. This patchy distribution of driver genes among unicellular eukaryotes, combined with the more common occurrence of driver states in those orthologs of driver genes that have been detected, might reflect the absence, in unicellular organisms, of some of the mechanisms that control cell division and cell–cell cooperation in multicellular life forms. These mechanisms appear to be shared by all multicellular life forms, even when they lack a multicellular common ancestor, and their failure results in cancer in metazoans. However, at present, this interpretation should be taken with caution because the relatively few protist genome sequences that are currently available poorly represent the enormous diversity of unicellular eukaryotes. Further analysis of the growing collection of protist genomes should clarify the links between drivers and multicellularity.

Despite the pronounced overall avoidance of the drivers, a sizable fraction of driver mutations appear as ancestral states across major clades including non-primate mammals. Although this might seem to provide evidence for 'molecular atavism'[58], for many drivers fixed at some point during species evolution, likely compensatory mutations were identified, and many more, probably, remain undetected. When available, examination of the corresponding protein structures often elucidates credible mechanisms by which the compensatory residue(s) could balance or counteract the effects of the driver through direct interaction (e.g., steric effects, pH, etc.) or modification of small molecule binding pockets or protein–protein interaction interfaces.

Here we employed a phylogenetics first approach to the identification of compensators which does not rely on structural information and, conversely, can inform subsequent structural

studies. As the body of available gene and protein expression data grows, this in silico approach for the identification of compensators can be augmented through validation of the functional effects of the drivers by utilizing transcriptome and proteome analyses. Separation of pairwise associations from noise in the MSA can be challenging[59–61], motivating the development of new methods[62]. Here we present a coherent approach to quantitatively assess relevance of such associations (see "Methods" for details). Achieving statistical significance requires a critical number of sequences to harbor the driver, which is unrealistic for extremely deleterious states, as well as a small ensemble of candidate compensators. For example, reviewing the well-characterized driver *PTPN11*: A72T[63], we identified a candidate compensator F285Y, which likely maintains interaction with the driver residue through hydrogen bonding, further supported by the observation that F285S is also a driver (Supplementary Fig. 19). However, notwithstanding this plausible biological argument, the probability of independent co-occurrence of the pair is high and does not pass our selection criteria. Thus, the conservative set of compensators we infer here is only a subset of all mutations compensating for the deleterious effects of drivers, in agreement with previous observations indicating that intra-protein epistasis is pervasive in evolution[64].

Previous work not only suggests the presence of many compensated missense mutations (even if the compensator is often unknown) across the species tree[65], with a long list for mice[66], but also that for every deleterious state, there are multiple, typically more than 10, possible compensatory mutations[67]. In the case of drivers, one would expect that the (putative) compensators detected in other species should be avoided in cancers, given that they mitigate the effect of drivers. As expected, we detected multiple compensators for many drivers, but surprisingly, we additionally found that numerous mutation pairs co-occurred at

much higher frequencies than expected by chance in both species and tumors (Fig. 5a). Such putative compensators were identified for the most commonly observed drivers (Supplementary Fig. 20). One could speculate that, in these cases, the compensation of the impairment of protein function caused by the driver mutation is only partial and results in a level of activity of the respective proteins that is optimal for tumor growth (put another way, certain uncompensated driver mutations could be deleterious even in tumors). Clearly, however, the causes of the seemingly paradoxical congruent associations between DM and compensatory mutations in tumors and in species evolution require further investigation. In particular, analysis of mutant allele frequencies (MAF) and examination of within-tumor selection signatures have the potential to demonstrate that driver MAFs are higher when paired with a compensator or otherwise clarify the underlying dynamics. Regardless of the underlying mechanism(s), these findings imply that many mutations that are considered to be drivers due to their repeated detection in tumors are actually compensators[68].

Altogether, our findings clearly indicate that most if not all cancer driver mutations are deleterious for the respective organisms irrespective of whether or not they are prone to cancer. For a substantial fraction of drivers, the deleterious effect is apparently so pronounced that they are universally avoided in evolution. However, the majority of the drivers appear as ancestral states in some groups of organisms, and for many of these, compensators are identifiable. Structural and functional investigation of the interactions between drivers and compensators can be expected to shed light on mechanisms of tumorigenesis; the roles of oncogenes and tumor suppressors in different organismal contexts; and protein evolution in general.

## Methods

**Construction of multiple sequence alignments (MSA)**. For each of the 84 driver genes considered, the sequence of the human gene (referred to as target sequence) was retrieved from the NCBI RefSeq database (see the following section on the differences between the RefSeq and COSMIC reference sequences). A single iteration of PSI-BLAST[69] was conducted against the Refseq database using default parameters, with the exception of no compositional adjustment, retrieving up to 10,000 database sequences. When close to 10k sequences were returned by the first PSI-BLAST iteration, and this list was almost exclusively metazoan, a second round of PSI-BLAST was conducted with the same parameters, but with Metazoa excluded from the search. These sequences were clustered and aligned as described previously[70]. Briefly, sequences are clustered with a similarity threshold of 0.5 and each cluster is aligned. Cluster-to-cluster self-score normalized similarity scores are then produced and clusters with a pairwise score >0.05 are aligned to each other. This step is performed iteratively.

The resulting clusters were examined for their taxonomic distribution and their alignments were manually compared in an effort to determine if the cluster containing the target sequence may be aligned with another cluster composed of complementary taxa. Upon this review, in all cases, the original target-containing cluster was retained without adding other sequences. Short sequences fragments were removed from the alignment. An approximate ML tree was generated from all alignments using FastTree[71], after filtered out sites with the with gap fractions >50% and homogeneity <0.1. This tree was manually reviewed for paralogs which were removed along with excess prokaryotic sequences when prokaryotes outnumbered eukaryotes. A new tree was generated with the remaining sequences in the same fashion and saved along with the full alignment including all positions containing at least one non-gap entry. The final tree was rooted on the taxonomically deepest internal branch.

**Differences between COSMIC references and Refseq entries**. In most cases, the reference sequence from the COSMIC "Mutation_AA" data matches a/the Refseq entry for the gene. For the following eight genes, the COSMIC Mutation_AA position data was modified to agree with Refseq by adjusting select COSMIC positions to account for the differences between the COSMIC reference sequence and the RefSeq entry. In the case of truncations, where the modified COSMIC Mutation_AA data still contain entries that do not correspond to the Refseq sequence, none of these positions harbor drivers, and some are inconsistently referenced within COSMIC (e.g., Mutation_AA data contains both R123G and G123W).

*CASP8*: pos 136-181, −32; pos 182-end, −17
*GNAS*: pos 1-end, +643

*KDM6A*: pos 445-end, +52
*PBRM1*: pos 1436-end, +52
*SETD2*: pos 1-end, +503
*SMARCA4*: pos 1393-end, +32
*STAG2*: pos 1157-end, +37
*TGFBR2*: pos 35-end, +25

The reconstructed COSMIC reference for *TCF12* does not agree with the Refseq sequence for *TCF12*, NP_003196.1, in the neighborhood around the top driver in this gene, C419Y, which could not be amended as described above and for this reason, the COSMIC reference sequence, Ensemble ID: ENST00000438423 was added to the alignment and used as the reference sequence instead.

**Homogeneity calculation**. Homogeneity values were calculated for each alignment column across vertebrate sequences as previously described in Yutin et al.[72]. Briefly, for each column two sum-of-pairs scores were calculated: (1) within the given column and a "homogenous" column with the same residue in all aligned sequences and (2) a column, composed of random amino acids. A linear scaling of these scores between 0 and 1 is reported as homogeneity[72]. "Neighborhood homogeneity" refers to the mean of the seven homogeneity values for the specified site and its six nearest neighbors.

**Leaf weight calculation**. Sequences were leaf-weighted as they appear in the tree according to the following protocol (Makarova et al.[73], Supplementary Fig. 21). First, the total tree weight is defined as the sum of all branch lengths. Then moving forward from root to leaves, a weight for each node is defined as the product of A, the sum of all branch lengths stemming from that node, and B, the weight of the preceding node, divided by C, the sum all branch lengths stemming from the previous node: $A*B/C$. This process is continued until the weights of all leaves are assigned.

**Ancestry estimation and compensatory ensemble construction**. After obtaining leaf weights, weighted character sets were constructed for every node following a modified form of the Fitch traceback algorithm[74] for each alignment column corresponding to a site in the human reference protein. The character weight vectors were constructed for each tree node with weights equal to the sum of the leaf weights, descending from this node and containing the given character. Then the weights in each vector were normalized to sum to 1. Next, pseudo-conditional probabilities were constructed for all interior nodes taken to be the normalized product of each vector with the vector assigned to its immediately ancestral node. Each node was then assigned the "consensus" residue with the highest weight if that weight was >0.5, or the "undefined" state otherwise. Transitions between residues were assigned to the midpoint of the branches, connecting nodes with different consensus residues. Compensatory ensembles were then constructed of states which transitioned along the same or a prior branch as the emergence of the driver in the ancestral record.

**Compensatory ensembles were held to meet the following criteria**.

(1) While there may be more than one unique compensatory mutation associated with each driver, all sequences containing the driver must also contain a member from this ensemble. This decreases the probability that a compensatory ensemble will be constructed for a driver additionally present in an arbitrary leaf of the tree due to sequencing error; however, drivers present in at least four leaves (the minimum required for subsequent steps) for which no compensatory ensemble was constructed tend to be more frequently observed in the MSA compared to those with a compensatory ensemble (Supplementary Fig. 22). This suggests that sequencing error is unlikely to be the cause of our inability to predict compensatory ensembles.

(2) The probability of this co-occurrence, assuming the presence of the putative compensatory mutation(s) and the driver are independent, is <1%. This probability was estimated to follow the binomial form:

$$\sum_{k=N_{\text{pair}}}^{N_{\text{total}}} \binom{N_{\text{total}}}{k} F^k (1-F)^{N_{\text{total}}-k}$$

where $N_{\text{total}}$ is the number of transitions to the driver state across the entire ancestral record, $N_{\text{pair}}$ is the number of transitions to the driver state which are descendent of a transition to a compensatory state, and $F$ is the fraction of the tree (fraction of all applicable branch lengths) occupied by a compensatory state.

(3) Members of the ensemble must appear at least twice independently, at two nodes in the tree.

(4) At least two sequences containing the driver must descend from each of these nodes.

(5) Each individual mutation may only occur in at most one ensemble. The ensembles were constructed in an order to minimize the probability of independent co-occurrence and ensembles 2 or more larger than the smallest ensemble for each driver are not discussed. A cartoon of a driver with a compensatory ensemble satisfying these criteria is shown in Supplementary Fig. 23.

**PDB structure scoring**. For each human reference protein, a single iteration of protein BLAST against the Protein Data Bank was conducted with default parameters, but retrieving up to 10,000 database sequences. For each hit, at every site, two relative score estimates were constructed, the "Global Score" (the BLAST score divided by the total protein length) and the "Local Score" (the BLAST score divided by the length of the footprint of the query sequence on the structure). For each site, the structure with the largest sum of the Local Score and Global Score squares was identified (Supplementary Fig. 7). The structure with the highest Local Score encompassing both sites of every driver/compensator pair was also recorded. The structures highlighted in Fig. 6 all have Local Scores of 1.9 or greater and coverage of driver/compensatory ensemble neighborhoods with structures scored 1.9 or greater is displayed in Fig. 5c.

**Calculation of association score for pairs of mutations**. For each gene, every pair of residues, with at least one being a driver state, that appear in at least one row in the MSA or one tumor the COSMIC database was processed as follows. For the COSMIC database, the observed number of pairs and expected number of pairs (the product of the frequency of residue 1, frequency of residue 2, and number of tumors in the dataset) was recorded. For the MSA, first the leaf weights were normalized so that the weights of sequences with non-gap residues in both sites of the pair sum to 1. Observed and expected values were then calculated as follows. The observed number was assigned the product of the sum of the weights corresponding to sequences harboring the pair and the number of nonzero-weighted sequences. The expected number was assigned the product of the sums of the weights corresponding to sequences harboring each state in the pair and the number of nonzero-weighted sequences. Note that this resulted in non-integer numbers of both expected and observed pairs in the MSA.

For all pairs of mutations, the association score (analogous to the log-odds ratio) was calculated as

$$\text{Score} = \begin{cases} -\ln(1 - \text{PCDF}(\exp, \text{obs})), & \text{obs} > \exp \\ \ln(\text{PCDF}(\exp, \text{obs})), & \text{obs} < \exp \end{cases}$$

where PCDF(exp, obs) is the cumulative probability of a Poisson distribution with mean "exp", the expected value of the data, and evaluated at "obs", the observed value of the data. Pairs with the highest scores in both the MSA and COSMIC databases are available in Tables S3 and S5. Scores with a magnitude of −1 to 1, indicating the pair that is observed about as frequently as it is expected, were discarded. Pairs with nonzero association scores in both COSMIC and the MSA are displayed in Fig. 5a.

**Statistics and reproducibility**. Regarding Fig. 2e: All distributions are suitably normal ($p < 0.01$, Anderson Darling) with significance ($p < 0.05$) reported according to a two-sample $t$-test.

**Reporting summary**. Further information on research design is available in the Nature Research Reporting Summary linked to this article.

## Data availability

All data utilized in this study is publicly available through the RefSeq, https://www.ncbi.nlm.nih.gov/refseq/, COSMIC, https://cancer.sanger.ac.uk/cosmic, and PDB, https://www.rcsb.org/, projects. The alignments generated for this work, from which all figures can be recreated, are made available at https://www.ftp.ncbi.nih.gov/pub/wolf/_suppl/drivers/.

## Code availability

All custom code designed for this study quantifying multisequence alignment and phylogenetic tree statistics is described in the supplementary materials in sufficient detail that implementation in the user's programming language of choice is possible. All custom code designed for this study, in addition to the protocol used to construct the multisequence alignments, is made available at https://www.ftp.ncbi.nih.gov/pub/wolf/_suppl/drivers/.

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

## Acknowledgements
We thank Koonin group members for helpful discussions. The authors' research is supported by funds of the Intramural Research Program of National Institutes of Health of the USA (National Library of Medicine).

## Author contributions
N.D.R., Y.I.W., and E.V.K. designed the study; N.D.R. performed research, N.D.R., Y.I.W., and E.V.K. conducted the data analysis; N.D.R. and E.V.K. wrote the manuscript that was edited and approved by all authors.

## Competing interests
The authors declare no competing interests.
