## [Peer Review File · Communications Biology]

This manuscript has been previously reviewed at another Nature Research journal. This document only contains reviewer comments and rebuttal letters for versions considered at Communications Biology.

Reviewers' comments:

Reviewer #1 (Remarks to the Author):

Thank you for taking the time to respond in detail to my comments and incorporate my suggestions into the revised manuscript. I think the biological insights come through much more clearly now in the updated version.

I have just one more request for revision. I think there was a bit of misunderstanding over my comment "It would be interesting to know whether drivers that co-occur with a compensator differ in their frequencies from those that do not".

By frequencies I meant mutant allele frequencies within individual tumours, not the numbers of tumours that had those mutations.

Mutant allele frequencies (MAFs) are important indicators of potential selection for or against certain mutations within a tumour sample.

If some drivers are indeed compensators I would expect their MAFs to be higher within samples where they co-occur with other drivers in the same protein than when they occur on their own.

Conversely, if compensators mitigate the deleterious effects of drivers, as speculated, then a similar trend would be seen.

However, if compensators reduce the survival advantages certain drivers confer to tumour cells, then such drivers should be found at lower frequencies when they co-occur with a compensator than when they do not.

This also represents a good 'sanity check' as well, because the frequencies of interacting drivers and compensators should track closely across samples and to rule out coincidental co-occurrence due to one of the mutations in the pair having low MAF.

I feel strongly the analyses are necessary to really support and illustrate the claims some drivers are actually compensators and that interactions between drivers and compensators affect tumour fitness.

Reviewer #2 (Remarks to the Author):

This study by Rochman et al describes a deep phylogenetic analyses of cancer driver mutations and potential compensatory mutations for these drivers. I'll note that this study leverages many bioinformatic, genomic and statistical methods for which I lack experience. Thus, I can really only judge it from the perspective that "if they're methods and bioinformatic results are sound", how reasonable and impactful their conclusions are. If so, then I would say that this manuscript is very interesting with significant implications for evolutionary biology (both organismal and somatic, with the latter leading to cancers). Their studies are phylogenetically much deeper than previous studies, allowing them to show that the presence of driver mutations in other species does not strongly relate to evolutionary distance from us. That most drivers are strongly avoided (in some cases, completely avoided) across other species is not surprising, but that a number of drivers are present and even ancestral in other taxa is fascinating. In particular, the case of KDM6A is striking, with the driver variants that are very common in cancers representing the ancestral state in most other reptiles and mammals (except for their absence in primates). Their analyses of compensatory mutations are also important, and assuming that their statistical methods are sound, should be impactful. As they note, one might expect to find such compensatory mutations accompanying cancer driver mutations in the germlines of other species, but that these

compensatory mutations are pretty common in cancers is surprising; as they note, this could represent a "tuning" for the optimal level of expression of the phenotype (a Goldilocks context). In fact, I would agree with their assessment that their results suggest that some if not many mutations considered as "drivers" may actually be compensatory mutations for the "real driver".

The authors seem to have addressed the comments of the two previous reviewers, again with the caveat that the methods used and interpretation of results are sound. Their explanations appear logical, and they have made changes within the text to better present their results.

In all, with the caveats stated above, I believe that this manuscript should make a significant impact on the cancer and evolutionary biology fields.

Minor but needs fixing:

- 1) In Table S1, the section "Notes on presentation within the tree" starting on page 15 appears jumbled and may just reflect a poor pdf conversion.
- 2) Table S7 has several blank pages and similarly looks poorly converted. Please check all supplemental tables.

Signed: James DeGregori

Dear Dr Koonin,

Your manuscript entitled "Deep Phylogeny of Cancer Drivers and Compensatory Mutations" has now been seen by 2 referees, 1 of which was an original reviewer of the earlier submission of the manuscript. You will see from their comments below that while they find your work of interest, some important points are raised. We are interested in the possibility of publishing your study in Communications Biology, but would like to consider your response to these concerns in the form of a revised manuscript before we make a final decision on publication.

Specifically, we ask that you perform an additional analysis as suggested by Reviewer 1 to determine whether drivers that co-occur with a compensator differ in their MAFs from those that do not.

While we agree with Reviewer 1 that this is a topic of interest, we explain below why our existing analysis is analogous to that requested and that calculating MAFs would require access to a different type of data which is outside the scope of this work. However, we believe we are able to address the additional points made by the reviewer with the existing analysis and so we have made the appropriate amendment to the text.

We therefore invite you to revise and resubmit your manuscript, taking into account the points raised. Please highlight all changes in the manuscript text file.

Best regards,

Caitlin Karniski, PhD
Associate Editor, Communications Biology
One New York Plaza, Suite 4600
New York, NY 10004-1562
orcid.org/0000-0002-1377-5118
caitlin.karniski@us.nature.com

Referee expertise:

Referee #1: Evolutionary cancer genomics

Referee #2: Cancer evolution

Reviewers' comments:

Reviewer #1 (Remarks to the Author):

Thank you for taking the time to respond in detail to my comments and incorporate my suggestions into the revised manuscript. I think the biological insights come through much more clearly now in the updated version.

We appreciate the detailed consideration and interest in this work.

I have just one more request for revision. I think there was a bit of misunderstanding over my comment "It would be interesting to know whether drivers that co-occur with a compensator differ in their frequencies from those that do not".

By frequencies I meant mutant allele frequencies within individual tumours, not the numbers of tumours that had those mutations.

Mutant allele frequencies (MAFs) are important indicators of potential selection for or against certain mutations within a tumour sample.

If some drivers are indeed compensators I would expect their MAFs to be higher within samples where they co-occur with other drivers in the same protein than when they occur on their own.

Conversely, if compensators mitigate the deleterious effects of drivers, as speculated, then a similar trend would be seen.

However, if compensators reduce the survival advantages certain drivers confer to tumour cells, then such drivers should be found at lower frequencies when they co-occur with a compensator than when they do not.

This also represents a good 'sanity check' as well, because the frequencies of interacting drivers and compensators should track closely across samples and to rule out coincidental co-occurrence due to one of the mutations in the pair having low MAF.

I feel strongly the analyses are necessary to really support and illustrate the claims some drivers are actually compensators and that interactions between drivers and compensators affect tumour fitness.

We regret any miscommunication. In the previous revision, we completed an additional analysis comparing driver frequency with respect to the presence of compensators *among* tumors rather than *within* tumors on the basis of data availability. All tumor sequencing data utilized in this study was retrieved from the COSMIC database where, to the best of our understanding, no data is available on within tumor MAFs.

First, it is important to clarify how MAFs are defined. To the authors' knowledge, MAF is typically defined to be either 1) $\text{mutant_reads}/(\text{mutant_reads}+\text{reference_reads})$ where reads may result from either single-cell or bulk sequencing and reference_reads come from adjacent normal tissue, or 2) $\text{mutant_reads}/\text{total_reads}$ where reads result from single-cell sequencing. While sequencing of normal tissue is rapidly increasing as is the number of tumors for which single-cell sequencing has been completed, to date, the vast majority of tumor sequence data results from bulk sequencing without a reference to adjacent normal tissue. While there might be a number of tumors included in the COSMIC database for which adjacent tissue or single cell data is available, this information is not readily accessible.

Calculating MAFs, in the strict sense above, would require access to additional datasets and entirely new, extensive analysis which we believe to be outside the scope of this work. However, in our view, the existing analysis is able to address most of the salient points raised by the reviewer:

Mutant allele frequencies (MAFs) are important indicators of potential selection for or against certain mutations within a tumour sample.

As described, we rely entirely on bulk tumor sequencing results. In this way, the presence or absence of a given mutation within a given tumor may be considered a binarization of the MAF distribution: in other words, presence corresponds to high MAF and absence to low MAF. The drivers considered in this work are selected on the basis of frequent observation among the bulk-sequenced tumor population; thus, by definition, they have high MAF among many tumors, i.e. evolve under positive selection.

If some drivers are indeed compensators I would expect their MAFs to be higher within samples where they co-occur with other drivers in the same protein than when they occur on their own.

Conversely, if compensators mitigate the deleterious effects of drivers, as speculated, then a similar trend would be seen.

In these cases, the presence of the compensator moves the MAF distribution to the right and increases the "high" population. This is precisely the observation presented in the supplementary figure added in the first round of revision: compensated drivers are more frequently observed than those which are not compensated.

However, if compensators reduce the survival advantages certain drivers confer to tumour cells, then such drivers should be found at lower frequencies when they co-occur with a compensator than when they do not.

In this case, one would expect the opposite result, that compensated drivers are less frequently observed.

This also represents a good 'sanity check' as well, because the frequencies of interacting drivers and compensators should track closely across samples and to rule out coincidental co-occurrence due to one of the mutations in the pair having low MAF.

We agree that greater statistical significance could be gained through the analysis of within-tumor selection signatures and believe this is an attractive avenue for future development. We expect noise attributable to common, low MAF mutations to be negligible given the considerations discussed above, namely, that mutations with low MAF are unlikely to appear in bulk sequencing.

I feel strongly the analyses are necessary to really support and illustrate the claims some drivers are actually compensators and that interactions between drivers and compensators affect tumour fitness.

We hope the reviewer finds these comments clarifying. We have amended the relevant portion of the Discussion to indicate that inclusion of single-cell sequencing has the potential to attain greater confidence regarding the proposed compensatory status of many drivers (ll. 518-521: "In particular, analysis of mutant allele frequencies (MAF) and examination of within-tumor selection signatures have the potential to demonstrate that driver MAFs are higher when paired with a compensator or otherwise clarify the underlying dynamics."

We additionally wish to emphasize that much of the existing analysis is devoted to contrasting the mutation frequency among tumors and among species. We view this comparison to be analogous to the first definition of MAF at the population level where the ensemble of reference species across the multicellular tree of life may be considered to take the place of “adjacent normal tissue”. We added a comment to that effect in the Discussion (ll. 446-447 “This analysis allows us to broadly assess the fitness effects of driver mutations across varying evolutionary spans.”)

Reviewer #2 (Remarks to the Author):

This study by Rochman et al describes a deep phylogenetic analyses of cancer driver mutations and potential compensatory mutations for these drivers. I'll note that this study leverages many bioinformatic, genomic and statistical methods for which I lack experience. Thus, I can really only judge it from the perspective that “if they're methods and bioinformatic results are sound”, how reasonable and impactful their conclusions are. If so, then I would say that this manuscript is very interesting with significant implications for evolutionary biology (both organismal and somatic, with the latter leading to cancers). Their studies are phylogenetically much deeper than previous studies, allowing them to show that the presence of driver mutations in other species does not strongly relate to evolutionary distance from us. That most drivers are strongly avoided (in some cases, completely avoided) across other species is not surprising, but that a number of drivers are present and even ancestral in other taxa is fascinating. In particular, the case of KDM6A is striking, with the driver variants that are very common in cancers representing the ancestral state in most other reptiles and mammals (except for their absence in primates). Their analyses of compensatory mutations are also important, and assuming that their statistical methods are sound, should be impactful. As they note, one might expect to find such compensatory mutations accompanying cancer driver mutations in the germlines of other species, but that these compensatory mutations are pretty common in cancers is surprising; as they note, this could represent a “tuning” for the optimal level of expression of the phenotype (a Goldilocks context). In fact, I would agree with their assessment that their results suggest that some if not many mutations considered as “drivers” may actually be compensatory mutations for the “real driver”.

We thank the reviewer for their interest and kind remarks. We can only hope future readers will have an equally positive assessment.

The authors seem to have addressed the comments of the two previous reviewers, again with the caveat that the methods used and interpretation of results are sound. Their explanations appear logical, and they have made changes within the text to better present their results.

In all, with the caveats stated above, I believe that this manuscript should make a significant impact on the cancer and evolutionary biology fields.

Again, we appreciate this positive assessment and hope and trust that the reviewer's assessment of the expected impact of this work is correct.

Minor but needs fixing:

- 1) In Table S1, the section “Notes on presentation within the tree” starting on page 15 appears jumbled and may just reflect a poor pdf conversion.
- 2) Table S7 has several blank pages and similarly looks poorly converted. Please check all supplemental tables.

We regret these formatting errors which have been amended.

Signed: James DeGregori

REVIEWERS' COMMENTS:

Reviewer #1 (Remarks to the Author):

To start I want to offer my apologies for taking so long to review the latest version of your manuscript.

I follow your reasoning in your rebuttal. I maintain that sufficient public bulk exome and WGS data exist to readily conduct the analysis I proposed. Even without matched normal it is possible to distinguish high frequency from low-frequency mutations, especially when one is focusing on specific mutations, rather than the collection of all mutations as a whole.

However, under the circumstances I can accept if you don't have time to complete this analysis. There are sufficient data already in the paper.

I appreciate the explanation you provided about the co-occurrence and drivers & compensators in the context of selection. That clarified in my mind the reasons for your interpretation of the outcomes of that analysis.

I would like to propose two further, minor changes to the manuscript to help readers better understand the logic and significance of the driver/compensator co-occurrence results.

First, a comment regarding the expected outcome of this analysis if there is selection for co-occurrence of driver with their compensators should be added, either in the Results or Discussion.

My second request regards the way mutation frequency is presented as a percentage in some graphs (Fig 2) but as a count in other graphs (Fig 4). It's not always clear what 'Frequency' means in each context.

In particular in Fig S20 it should be made clear that Frequency means the number of tumours where a given mutation was observed, by changing the labels on the y axes and providing this explanation in the legend.

In general, it would be good to amend all figure legends to clarify whether Frequency means percentage or overall count.

Once these changes are made, I'll be happy for this paper to be accepted for publication.

Reviewers' comments:

To start I want to offer my apologies for taking so long to review the latest version of your manuscript.

I follow your reasoning in your rebuttal. I maintain that sufficient public bulk exome and WGS data exist to readily conduct the analysis I proposed. Even without matched normal it is possible to distinguish high frequency from low-frequency mutations, especially when one is focusing on specific mutations, rather than the collection of all mutations as a whole.

However, under the circumstances I can accept if you don't have time to complete this analysis. There are sufficient data already in the paper.

I appreciate the explanation you provided about the co-occurrence and drivers & compensators in the context of selection. That clarified in my mind the reasons for your interpretation of the outcomes of that analysis.

I would like to propose two further, minor changes to the manuscript to help readers better understand the logic and significance of the driver/compensator co-occurrence results.

First, a comment regarding the expected outcome of this analysis if there is selection for co-occurrence of driver with their compensators should be added, either in the Results or Discussion.

My second request regards the way mutation frequency is presented as a percentage in some graphs (Fig 2) but as a count in other graphs (Fig 4). It's not always clear what 'Frequency' means in each context.

In particular in Fig S20 it should be made clear that Frequency means the number of tumours where a given mutation was observed, by changing the labels on the y axes and providing this explanation in the legend.

In general, it would be good to amend all figure legends to clarify whether Frequency means percentage or overall count.

Once these changes are made, I'll be happy for this paper to be accepted for publication.

Response:

With regard to the reviewer's comment on selection for co-occurrence of driver with their compensators, we added the following to the Discussion:

“Such putative compensators were identified for the most commonly observed drivers (Supplementary Figure 20). One could speculate that, in these cases, the compensation of the impairment of protein function caused by the driver mutation is only partial and results in a level of activity of the respective proteins that is optimal for tumor growth (put another way, certain uncompensated driver mutations could be deleterious even in

tumors). Clearly, however, the causes of the seemingly paradoxical congruent associations between DM and compensatory mutations in tumors and in species evolution require further investigation.”

We believe that this discussion point addresses the issue to the extent currently possible.

In the revised version of the manuscript, we indicate, wherever appropriate, that Frequency refers to fraction.